# Organic carbon burial in global lakes and reservoirs

Raquel Mendonça[1,2], Roger A. Müller [1], David Clow[3], Charles Verpoorter[1,4], Peter Raymond[5], Lars J. Tranvik [1] & Sebastian Sobek[1]

Burial in sediments removes organic carbon (OC) from the short-term biosphere-atmosphere carbon (C) cycle, and therefore prevents greenhouse gas production in natural systems. Although OC burial in lakes and reservoirs is faster than in the ocean, the magnitude of inland water OC burial is not well constrained. Here we generate the first global-scale and regionally resolved estimate of modern OC burial in lakes and reservoirs, deriving from a comprehensive compilation of literature data. We coupled statistical models to inland water area inventories to estimate a yearly OC burial of 0.15 (range, 0.06–0.25) Pg C, of which ~40% is stored in reservoirs. Relatively higher OC burial rates are predicted for warm and dry regions. While we report lower burial than previously estimated, lake and reservoir OC burial corresponded to ~20% of their C emissions, making them an important C sink that is likely to increase with eutrophication and river damming.

---

[1] Limnology/Department of Ecology and Genetics, Uppsala University, 75236 Uppsala, Sweden. [2] Department of Biology, Federal University of Juiz de Fora, 36036-330 Juiz de Fora, Brazil. [3] U.S. Geological Survey, Colorado Water Science Center, Denver, CO 80225, USA. [4] Univ. Littoral Cote d'Opale, Univ. Lille, CNRS, UMR 8187, Laboratoire d'Océanologie et de Géosciences, F 62930 Wimereux, France. [5] Yale School of Forestry and Environmental Studies, New Haven, CT 06511, USA. Correspondence and requests for materials should be addressed to R.Mça. (email: fm.raquel@yahoo.com.br)

Approximately one half of the total terrestrial carbon (C) pool transported by inland waters reaches the sea, due to large amounts of C being both emitted to the atmosphere and sequestered in sediments[1–3]. Carbon gases (carbon dioxide and methane) emitted from inland waters to the atmosphere act as greenhouse gases but are recycled in the biosphere on contemporary time scales, while C stored in sediments enters the long-term geological cycle. Studies at local and regional scales have shown that even if the C burial flux in lakes and reservoirs is often small compared to C emission, it represents a significant long-term C sink[4–7]. In addition, inland waters are more efficiently burying C than oceans, since a higher fraction of settling organic carbon (OC) escapes mineralization and stays in the sediments. This is due to higher sedimentation rates, lower oxygen availability, and higher proportion of land-derived OC in inland waters[8]. Recent studies have also indicated that inland water OC burial rates have been increasing over the Anthropocene due to soil erosion, river damming, and eutrophication[6, 9, 10].

Even though the importance and efficiency of lake and reservoir sediment C burial is widely recognized, global estimates are not well constrained. While the entire oceanic sediment sequesters ~0.2 Pg C per year (IPCC, 2013), global estimates of inland water OC burial range from 0.2 to 1.6 Pg C per year[1–3,7,11–14]. Even though some of the discrepancy between these estimates arises from differences in environments that were considered (lakes, reservoirs, floodplains, wetlands, and colluvium), there is considerable uncertainty in estimates even when the same environments were studied. The lack of a robust global estimate of inland water OC burial is largely due to a lack of measurements, which is related to the complexity of methods and the heterogeneity of sedimentation. Global burial estimates rely typically on few data of limited geographical distribution, and many assumptions. The question of how much C is sequestered in global lakes and reservoirs, then, remains open even though the number of studies including direct OC burial measurements has increased over the last decade.

To provide a better-constrained estimate of this fundamental and significant term of the continental carbon balance, we have compiled modern (last ~150 years) whole-basin OC burial data from the literature. We performed upscaling using multiple scenarios consisting of different predictive equations from OC burial models, and different inventories of inland water area, thereby providing a range of estimates (0.06–0.25 Pg C per year) that is intended to reflect the uncertainty caused by data scarcity.

## Results

**Modern OC burial measurements.** Direct in situ measurements of OC burial in inland waters are not only scarce, but also unevenly distributed globally. We gathered values from 403 different lakes and reservoirs which represent, for example, only 5% of the number of systems for which dissolved carbon dioxide concentration data are available (7939 lakes and reservoirs[15]). Lakes and reservoirs in North American and European countries account for ~90% of the water bodies in our data set while many countries, including some rich in inland waters (e.g., South American countries, Russia, Indonesia, India, Bangladesh, Vietnam, and Congo) have very few or no direct measurements. Not a single OC burial measurement was registered in 85% of the world's COSCATs (major catchments based on a coastal segmentation and related catchments analysis[16]) (Supplementary Fig. 1).

Modern, whole-system OC burial rates varied from 0.2 to 17,392 g C m$^{-2}$ yr$^{-1}$ (average of 250 and median of 40 g C m$^{-2}$ yr$^{-1}$; note that rates are expressed relative to the water body area) with higher values in reservoirs (median of 291 g

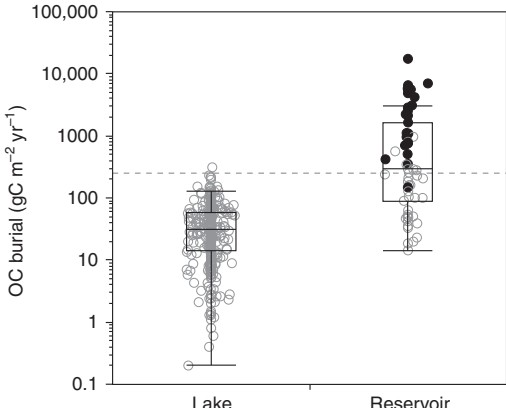

**Fig. 1** Modern organic carbon burial rates from global lakes and reservoirs. Each circle corresponds to one sampled system; small agriculture ponds from ref. [27] are in black. Box plots show median (line), interquartile range (box) and 10th–90th percentile (whiskers). The horizontal dashed line indicates global average. $n = 344$ for lakes and 59 for reservoirs

C m$^{-2}$ yr$^{-1}$), particularly in small agricultural ponds (Fig. 1). Artificial reservoirs have previously been shown to bury carbon at higher rates than natural lakes, owing to higher sedimentation rates and better condition for OC preservation (e.g., sediment anoxia, refs. [7, 8, 17,]). Because of this, reservoirs were treated separately from lakes in this study.

**Upscaling from local to global OC burial.** Upscaling was performed by multiplying OC burial rates with global lake and reservoir areas. In order to account for the scarcity of published lake and reservoir OC burial data, as well as for the uncertainty in lake and reservoir global areas, we calculated lake and reservoir OC burial based on four different scenarios, which result from two different approaches to model OC burial rates, and two different estimates of global lake and reservoir area. The range of values obtained from the different scenarios therefore reflects the sensitivity of estimated global OC burial to the key input parameters.

To build predictive models of OC burial, the watershed of each lake and reservoir in our data set was identified through the WWF HydroBASINS tool (http://www.hydrosheds.org) and their watershed characteristics were extracted (Methods section). Geostatistical models to predict OC burial rates in the world's COSCATs were developed using stepwise multiple linear regression (MLR), including as potential explanatory variables the watershed characteristics, system type (lake or reservoir) and system area. The MLR indicated (Supplementary Table 1) that the areal proportion of cropland, temperature and runoff in the catchments have positive influence on burial, while lake/reservoir area and average slope of terrain in the catchments have negative influence. These relationships are related to the stock, mobilization, and transport of OC from catchments, and the size of the receiving sedimentary basin. In addition, the positive relationships of OC burial with areal proportion of cropland and temperature may point toward a link between aquatic and terrestrial watershed productivity and OC burial. The negative effect of slope may be related to the fact that steep terrain is typically characterized by higher altitude and lower productivity, which may imply lower catchment OC export. Temperature was a significant predictor of OC burial, similar to what was found in an assessment of lakes and reservoirs in the USA[18], but the relationship was only strong in systems situated in COSCATS with an annual mean air temperature ≲15 °C, and not significant

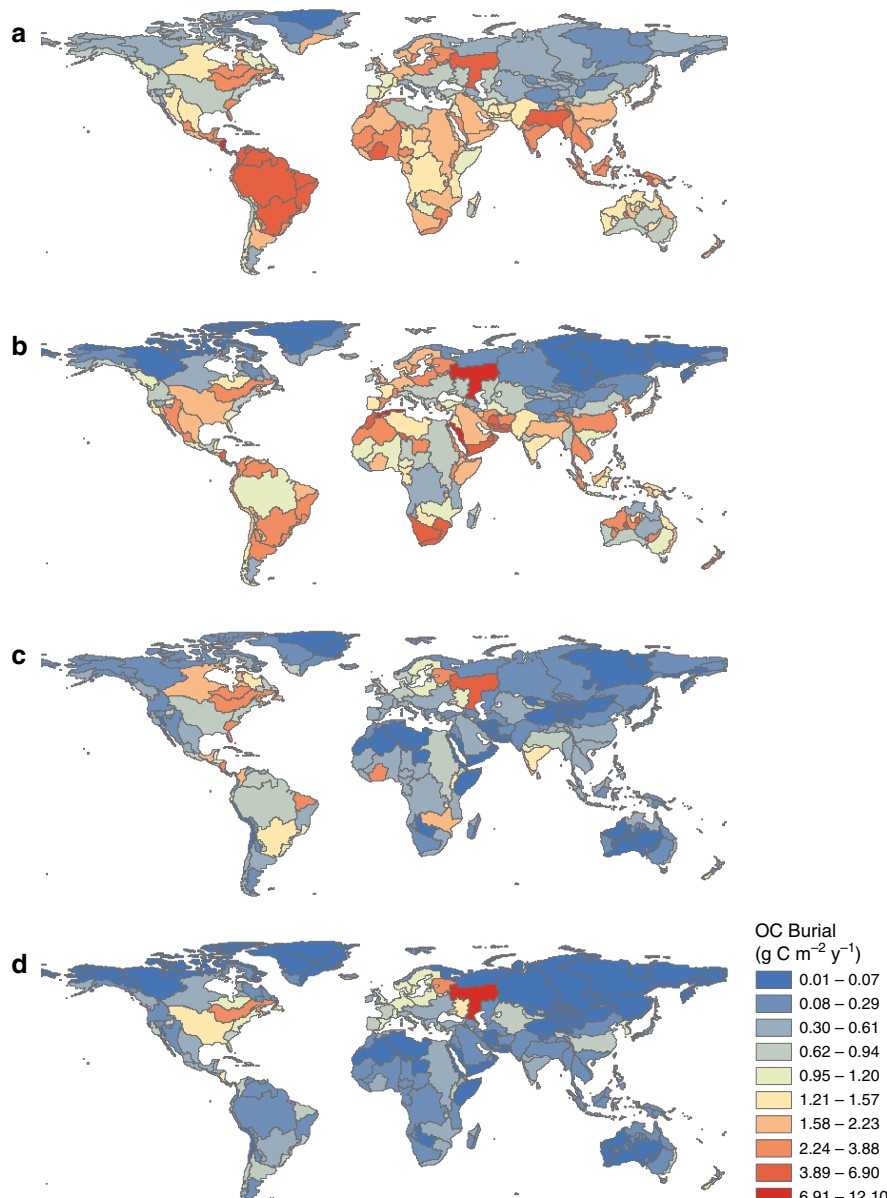

**Fig. 2** Organic carbon burial rate per total COSCAT area. The rates were calculated by dividing the total organic carbon burial in a COSCAT, in g C per year, per the total COSCAT area. **a** Scenario AG, one model for all data, GLOWABO[20] areas; **b** scenario SG, model for cold COSCATs, average burial for warm COSCATs, GLOWABO areas; **c** scenario AR, one model for all data, Raymond et al.[15] areas; **d** scenario SR, model for cold COSCATs, average burial for warm COSCATs, Raymond et al.[15] areas

in systems situated in COSCATS $\gtrsim$15 °C (Supplementary Fig. 2). The relationship between OC burial and temperature (Supplementary Fig. 2) indicates that the regression model including systems from all climate zones (method A, all: model 1, $n = 362$, $R^2$adj = 0.51, $p < 0.0001$, Supplementary Fig. 3a, Supplementary Table 1) may overestimate burial in warm lakes. To account for the apparently nonlinear effect of temperature on OC burial, we used a second approach (method S, split), in which we estimated burial in COSCATs <15 °C from a model including temperature (model 2, $n = 334$, $R^2$adj = 0.57, $p < 0.0001$, Supplementary Fig. 3b, Supplementary Table 1), and in COSCATs >15 °C from median values of lakes and reservoirs of different size classes (Methods section). No model of acceptable predictive power could be fitted to lakes and reservoirs in >15 °C COSCATs, probably because of the scarcity of data in the tropics ($n = 34$).

These two methods (A, all and S, split) were then used in combination with two different inventories of global inland water area. The first was adapted from the GLWD data set (Global Lakes and Wetlands Database[19] for lakes and reservoirs by Raymond et al.[15]) and utilizes size distribution relationships from the literature to statistically estimate the area and abundance of small lakes and reservoirs (inventory R). The second inventory is based on high-resolution satellite images and utilizes algorithms for inland water body detection (GLOWABO data set, ref. [20], inventory G) (Supplementary Fig. 4). Based on the scaling methods (A, S) and lake area inventories (R, G) described here, we produced the global modern OC burial scenarios AR, AG, SR, SG (Fig. 2, Table 1).

**Global OC burial in lakes and reservoirs.** Global OC burial rates resulting from the four scenarios varied from 19 to 48 (average 33) g C m$^{-2}$ inland water area yr$^{-1}$ and from 0.4 to 1.9 (average 1.1) g C m$^{-2}$ COSCAT area yr$^{-1}$. Globally, lake and reservoir OC

**Table 1 Summary of global estimates derived from the four scenarios and their mean**

|  |  | AG | SG | AR | SR | Mean |
|---|---|---|---|---|---|---|
| Total area (km$^2$) | Lakes | 4,799,573 |  | 2,739,766 |  | 3,769,669 |
|  | Reservoirs | 446,824 |  | 261,243 |  | 354,033 |
|  | Total | 5,246,396 |  | 3,001,009 |  | 4,123,702 |
| OC burial rate per water body area (g C m$^{-2}$ yr$^{-1}$) | Lakes | 37 | 20 | 19 | 14 | 22 |
|  | Reservoirs | 165 | 239 | 109 | 63 | 144 |
|  | Average, total[a] | 48 | 39 | 26 | 19 | 33 |
| OC burial rate per continental area (g C m$^{-2}$ yr$^{-1}$) | Lakes | 1.32 | 0.72 | 0.38 | 0.30 | 0.68 |
|  | Reservoirs | 0.55 | 0.80 | 0.21 | 0.12 | 0.42 |
|  | Total | 1.88 | 1.52 | 0.59 | 0.42 | 1.10 |
| Total OC burial rate (Pg C per year) | Lakes | 0.18 | 0.10 | 0.05 | 0.04 | 0.09 |
|  | Reservoirs | 0.07 | 0.11 | 0.03 | 0.02 | 0.06 |
|  | Total | 0.25 | 0.20 | 0.08 | 0.06 | 0.15 |

AG, scenario from method A (all data modeled) and inventory G (GLOWABO[20]) for inland water area; SG, scenario from method S (<15 °C modeled, >15 °C using median) and inventory G; AR, scenario from method A and inventory R (from Raymond et al.[15]) for inland water area; SR, scenario from method S and inventory R
[a]Calculated as the weighted average, considering the fluxes and the area occupied by lakes and reservoirs in each COSCAT

burial is estimated to range between 0.06 and 0.25 (average 0.15) Pg C per year (Table 1). Considering the mean of the four scenarios, OC burial rates per unit of continental area in the world's geographic zones (as in Supplementary Fig. 1) are: 0.77 g C m$^{-2}$ yr$^{-1}$ in the southern temperate zone; 1.28 in the southern subtropical zone; 1.35 in the tropical zone; 1.10 in the northern subtropical zone; 0.88 in the northern temperate zone; and 0.16 in the northern polar zone. Large variability in OC burial rates is observed within each geographic zone (Fig. 2) due to the scarcity and uneven distribution of OC burial measurements (Supplementary Fig. 1) and to the relatively patchy distribution of inland waters (Supplementary Fig. 4).

The G scenarios (using GLOWABO areas) result in higher values, mainly due to the ~80% larger total area of small inland water systems (0.001–1 km$^2$), where carbon is buried at higher rates, when compared to the R scenarios (areas as in ref. [15]). As compared to A scenarios (all data modeled), S scenarios (<15 °C modeled, >15 °C using median) result in lower burial rates, except for reservoirs, when GLOWABO areas are used (Table 1). The four scenarios are more divergent in warmer and drier COSCATs (Supplementary Fig. 5), where more direct measurements are urgently needed. Scenarios AG and SG show very high OC burial in dry regions (North Africa, Arabian Peninsula, and Australia), indicating that GLOWABO may overestimate water presence in these areas (Supplementary Fig. 4), probably due to a shadowing effect. The scenarios derived from Raymond et al.[15] inventory (AR and SR) give lower estimates in deserts, but are likely to underestimate OC burial globally due to under-representation of small systems. In addition, the Raymond et al.[15] data set includes only lakes and reservoirs while GLOWABO includes large rivers as well, which however is of limited influence for the global estimates since the area of the 5 largest stream orders combined was estimated at only 4% of the global lake and reservoir area.

Our highest global OC burial estimate, which is the first based on a global-scale, regionally resolved geostatistical analysis, is at the lower end in the range of previous global estimates (0.2–1.6 Pg C per year[1–3, 7, 11–14]. A similar result was obtained by a regional assessment[6], which concluded that OC burial in European lakes is smaller than previously estimated. Part of this divergence is due to the fact that some estimates[2, 12, 13] account for C burial in wetlands, floodplains, alluvial, and colluvial sediments, while our estimate only refers to lake and reservoir burial. We conclude that lake and reservoir C burial is estimated to be similar to ocean C burial (0.15 Pg C per year in lakes and reservoirs and 0.2 Pg C per year in the ocean), demonstrating that

inland water OC burial is greater on an areal basis. In addition, the burial of 0.15 Pg C per year in inland waters (mean of our scenarios) represents a removal of 0.3% (range of 0.1–0.5% for the four scenarios) of global terrestrial net primary production (~52 Pg C per year, estimate for 1990–2009, IPCC) to sinks operating at time scales of decades-centuries (reservoirs) to millennia (lakes). Furthermore, our estimated lake and reservoir burial rate was similar to the estimated net global carbon flux of soils, which is negative in the perturbed C cycle (net carbon loss of ~0.15 Pg C per year) due to increased erosion[21].

Anthropogenic increase in erosion and aquatic productivity (eutrophication), together with river damming, are responsible for a large increase in OC burial since pre-industrial times[9, 10, 12, 14, 22], estimated as ~0.05 Pg C per year[21] or 33% of our average OC burial. Similarly, recent average carbon accumulation rates in European lakes were twice as high as compared to the mean accumulation rate over the Holocene[6]. Despite the increase in soil erosion due to anthropogenic land cover change, sediment delivery to the sea has decreased by 15% compared to pre-Anthropocene values owing to the proliferation of dams[23, 24]. Efficient sediment trapping by dams combined with frequently low dissolved oxygen in bottom waters has been related to reservoirs in general accumulating OC at higher rates than natural lakes[17, 25]. Indeed, our scenarios result on average six times (4–12, range of scenarios) higher OC burial rates per unit of area in reservoirs than in natural lakes. Consequently, total burial in reservoir sediments (0.06 Pg C per year, mean of the four scenarios) amounts to ~60% of OC burial in natural lakes, even though lakes occupy a ~10 times larger total area (Table 1). In this context, it is important to note that the extent to which land-derived OC burial in reservoir sediments can be accounted as a new sink is not known; if the OC originated from land, it may also have been buried elsewhere (e.g., in floodplains or in the ocean) in the absence of the dams and, therefore, simply represents a change in storage location but not a new sink[26]. Some studies indicate, though, that the burial of land-derived OC may be more efficient in reservoirs than in other depositional environments[17, 25]. Thus, the additional fraction of land-derived OC that escapes mineralization because it is buried in a reservoir, and not in another depositional environment, may be accountable as a new, and anthropogenic, sink of land-derived OC. It is worth noting that this argument also applies to the burial of land-derived OC in natural lakes, which may only be counted as a new sink with regard to that fraction of OC that is more efficiently preserved in inland water sediments than in soils. Independently of this reasoning, a large share of the present-day annual inland

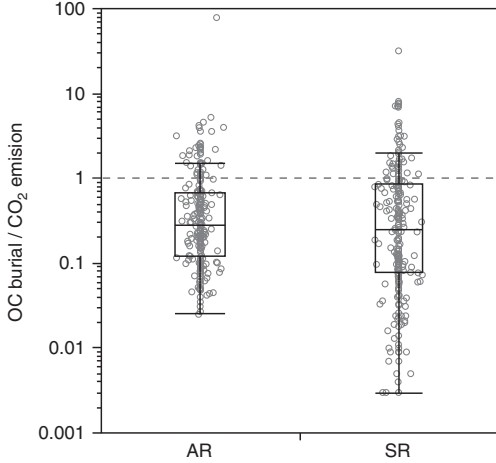

**Fig. 3** OC burial compared to $CO_2$ emission in lakes and reservoirs in global COSCATs. The burial/emission are derived from the organic carbon (OC) burial scenarios AR (method A, inventory R, $n = 230$) and SR (method S, inventory R, $n = 230$), where values above 1 (horizontal dashed line) indicate OC burial > $CO_2$ emission. Each circle corresponds to one COSCAT. Box plots show median (line), interquartile range (box) and 10th–90th percentile (whiskers)

water OC burial takes place in reservoirs, and this calls for the development of strategies in order to protect the accumulated carbon stock at dam decommissioning.

Eutrophication, i.e., high aquatic productivity in response to anthropogenic nutrient increase, is probably tightly linked to high OC burial rates. For example, the extremely high OC burial in small agricultural ponds and reservoirs has been attributed to their highly eutrophic state[27] (Fig. 1). Intensive agriculture is the main source of soil erosion with consequent transport of sediments, terrestrial OC and nutrients to inland waters[28, 29], and may therefore enhance OC burial both directly, through the high delivery and effective preservation of terrestrially derived OC, and indirectly, though nutrient enrichment (eutrophication) stimulating aquatic productivity and, thus, the sedimentation of aquatic OC[10, 22, 30].

The burial of autochthonous OC (derived from aquatic carbon dioxide fixation) can be regarded as a new C sink that has to be counted in continental C budgets. However, care must be taken to avoid double accounting of aquatic OC burial when establishing mass balances, since carbon dioxide ($CO_2$) uptake by photosynthesis is already accounted within net water-atmosphere $CO_2$ exchange. Further, only a small (but also unknown) fraction of the $CO_2$ fixed by aquatic primary production tends to become buried in the sediment, since aquatic organic matter is easily decomposed compared to land-derived OC, even in anoxic sediments[8, 31]. In addition, high degradation rates of aquatic OC lead to high methane ($CH_4$) production in eutrophic sediments, and a certain proportion of the $CH_4$ escapes the oxidation to $CO_2$ by aquatic microbes and reaches the atmosphere[32–34]. Considering that $CH_4$ has a 34-fold higher warming potential than the $CO_2$ taken up by photosynthesis (over a time interval of 100 years; IPCC 2013), the C processing in the sediments of eutrophic freshwater might result in a positive net effect on radiative forcing at a centennial time scale, in spite of considerable rates of sediment OC burial[26].

From a mass balance point of view, our analysis shows that comparable amounts of C are buried in sediments and emitted as gas from lakes and reservoirs worldwide. This comparison is based on $CO_2$ emissions only, since $CH_4$ emission is negligible in terms of C units. Our average OC burial (from R scenarios only, in order to allow comparison: 0.06–0.08 Pg C per year) represents

~20% of the estimated total $CO_2$ emission from lakes and reservoirs (0.32 Pg C per year[15]). However, burial exceeds $CO_2$ emission rates per water body area in some of the COSCATs (Fig. 3), most of them situated in dry regions (i.e., with low percentage inland water area in relation to COSCAT area; Supplementary Fig. 6a). This is possibly due to high soil erosion rates and to the deposition of terrestrial sediment load in few and small sedimentary basins causing higher OC burial rates; clearly, the high OC burial/$CO_2$ emission ratio in dry COSCATS was not related to exceptionally low $CO_2$ emission in dry regions (Supplementary Fig. 7c). Burial/emission ratios also increase with increasing the proportion of reservoir area (as related to total lake and reservoir area, Supplementary Fig. 6b), owing to the strong positive effect of dams on OC burial, while $CO_2$ emissions are not affected (Supplementary Fig. 7b), except for the first ~15 years after impoundment[35]. There was a weak relationship between OC burial and $CO_2$ emission rates per unit of water body area, even though the COSCATs standing out for the highest $CO_2$ emissions were also in the upper end of the range of OC burial rates (Supplementary Fig. 8). Almost all regions showing high burial and high emission per unit of water body area are located in the tropical zone (Supplementary Fig. 9), which suggests that high productivity turns inland waters into particularly strong carbon processors. We also point out that the variability in $CO_2$ emission between COSCATs is much smaller than the variability in OC burial (based on the range of values, Supplementary Fig. 8), illustrating that more research is needed on burial.

Our results endorse burial as a substantial component of inland water carbon cycling[1, 3] although it is noteworthy that C burial cannot compensate for a corresponding amount of C emission. Even though C burial per unit of area in lakes and reservoirs apparently is more effective than in soils[21], the degree to which inland water OC burial is accountable as a new C sink is presently unclear. More knowledge on the relative contribution of terrestrial and aquatic OC to sediment OC burial, and on the fates of these OC sources, is crucial to refine carbon budgets of the continents.

Our analysis points toward strategies to reduce uncertainties in future estimates of global OC burial. More systematic studies at lower latitudes could reduce data scarcity in warmer and drier regions, where the highest OC burial rates were predicted. Burial assessments would also profit from further investigations in reservoirs and eutrophic systems, given their significant contribution to global burial rates, and due to the likelihood that these systems will become even more abundant in the future. Also, the mapping of agricultural ponds, which tend to be highly eutrophic, would improve OC burial upscaling. Importantly, our estimates do not include floodplains and other near-water landscapes, which represent poorly constrained but most probably significant sites of carbon processing[36]. The uncertainties in upscaling to global carbon fluxes also decrease as the methods for counting and measuring global inland water area develop. New inventories of inland water area (e.g., Hydro-LAKES[37], GIW v1.0[38], and Pekel's time-resolved data set[39]) will allow for further improvements of inland water carbon fluxes estimates. However, even though new measurements and more accurate inventories undoubtedly would help to calibrate or update our estimates, the results presented here clearly demonstrate that OC burial in lakes and reservoirs is smaller, yet not less important, than previously estimated.

## Methods

**OC burial data compilation**. Our data set was composed of 475 observations of modern (i.e., the last ~150 years) OC burial rates, mainly derived from [210]Pb- and [137]Cs-dated cores, in 403 different lakes and reservoirs. Our analysis relied exclusively on whole-system OC burial rates (i.e., a single spatially resolved rate for each system), thus we treated the observations as follows. If data on whole-system

OC burial from the same system were available in different publications, we used the average value. If OC burial corrected for sediment focusing was estimated at two different sub-basins of a lake, we considered the average rate of the different zones as the whole-system OC burial. If data come from a single core sampled in the deepest zone of a system, we applied a correction for sediment focusing, which is the tendency for preferential deposition of sediment in the deepest part of a lake[40]. First, we split the lakes for which the sediment focusing factor (SFF) was reported into groups according to their maximum depth (≤5 m, >5–10 m, >10–30 m, >30–90 m, and >90 m); this approach assumes that mean lake bed slope is higher in deeper systems, and that SFF is positively related to mean lake bed slope[1]. We were not able to group systems by mean lake slope, since bathymetric data were not available for all systems. We then used the average SFF from each group to correct for sediment focusing the rates of other lakes within the same range of maximum depth. Two reservoirs in our data set (IDs 209 and 208 in Supplementary Data) which were formed by river dams and for which a single core was taken at their deepest zone (dam area) were not corrected for focusing, since sediment tends to accumulate at higher rates at river inflow areas[41]. If data come from single cores sampled at two or more sites (at different water column depths) in a system we took the average of the OC burial in the sites, considering that they would be representative of the spatial variability within that system. This approach was applied for 4 lakes/reservoirs—Lake Wohlen (8 sites), Lake Kinneret (4 sites), Lake Constance (4 sites), and Lake Brienz (2 sites)—all from Sobek et al.[8]. Even if these sites may not fully represent sediment accumulation in these basins, potential uncertainties have a small effect on the overall analysis, given that this approach was applied on 4 systems only, and that none of these 4 systems deviated statistically from the rest of the data.

We applied data exclusion criteria when searching the literature. We excluded data when the rates encompassed only long-term (Holocene-scale) OC burial; when the methods were obscure or in references that we could not find, such that the time scale of the reported OC burial rates could not be determined; and when OC burial data were presented in graphs only, with no details on systems name and geographic coordinates, and these details could not be retrieved in any other way.

**Watershed extents and characteristics**. The watershed extent of each lake and reservoir in our data set was identified using the WWF HydroBASINS tool (http://www.hydrosheds.org). Then, the following watershed characteristics were extracted to all lake and reservoir watersheds and to the COSCATs: watershed area (km$^2$), average watershed slope (degrees), average altitude (m), land cover (% of each land cover class), average annual temperature (°C), average soil OC at 1 m depth (kg C m$^{-2}$), average net primary productivity (g C m$^{-2}$ y$^{-1}$), total annual precipitation (mm), and average annual runoff (mm per year). Data on watershed slope and altitude were obtained from digital terrain maps of 250-m resolution as generated from the SRTM of 90-m resolution produced by the National Aeronautics and Space Administration (NASA, http://www.cgiar-csi.org). Land cover data were derived from maps of 1000-m resolution (Global Land Cover Project, GLC2000), made available by the European Commission's science and knowledge service, including 23 land cover classes (http://forobs.jrc.ec.europa.eu/products/glc2000/glc2000.php). Data on annual temperature, soil OC and net primary productivity were obtained from the Atlas of the Biosphere (https://nelson.wisc.edu/sage/data-and-models/atlas/). Finally, runoff data were derived from UNH/GRDC Composite Runoff Fields (http://www.grdc.sr.unh.edu).

These geographic analyzes were possible for systems within the HydroBasins tool coverage (below 60° northern latitude) and for which geographic coordinates were available. All geographic analyzes were performed in the software package ArcGIS 10.2 (ESRI).

**Statistical modeling**. In the predictive models, we only used lakes and reservoirs for which we could determine the watershed extents and characteristics (total of 368 systems). Lakes and reservoirs displayed systematic difference in OC burial rates, and were therefore treated as categorical predictors in modeling. We considered as reservoirs all types of constructed impoundments, e.g., hydropower and water supply reservoirs, as well as agricultural and urban ponds. The number of reservoirs in our data set was small (59 systems) compared to natural lakes.

Geostatistical models to predict OC burial rates in each COSCAT were developed using stepwise multiple linear regression (MLR). The potential explanatory variables were the watershed characteristics listed above, system type (lake or reservoir) and surface area. We measured surface area by analyzing satellite images in cases when this information was not available in the literature. Skewed variables were transformed by decadal logarithm to approach normality. We used forward stepwise MLR, and the minimum Akaike Information Criterion (AICc) for model selection.

We are aware that some system-specific variables, such as productivity, bottom water dissolved oxygen concentration, average or maximum depth, and water retention time, are probably important predictors of OC burial. However, they cannot be used for upscaling to COSCATs at present due to insufficient information, so we did not include these variables in the models.

The statistical analyzes were performed using JMP 11.0 (SAS Institute, USA).

**Global lake and reservoir area**. Our upscaling was based on two inland water area inventories—inventory R, from ref. [15] and inventory G, GLOWABO, from ref. [20]. Both data sets were clustered by COSCAT zones and we calculated the total area of water bodies for each of the following size classes—≥0.001–0.1 km$^2$, >0.1–1 km$^2$, >1–10 km$^2$, >10–100 km$^2$, >100 km$^2$. Since inventory R lists lakes and reservoirs separately, we used its lake/reservoir ratio from each size class in each COSCAT to estimate reservoir area in inventory G, which does not identify water bodies as either lakes or reservoirs.

**Upscaling**. Global OC burial was estimated after modeling OC burial in each of the world's COSCATs, using the MLR predictive equations and the total area of lakes and reservoirs in each COSCAT. In order to account for surface area (one of the MLR predictors), we used the median surface area of each size class range (e.g., 5.5 km$^2$ was the median surface area of the area size class of 1–10 km$^2$). Thus, we modeled the OC burial rates separately for each size class in each COSCAT. As system type (lake or reservoir) was also a predictor in MLR, we estimated OC burial separately for lakes and reservoirs within each size class range and in each COSCAT. Then, we estimated total COSCAT-wide OC burial rates from the weighted mean OC burial (considering the contribution of each system type and size class to the total area of lakes and reservoirs). OC burial rates were expressed as g C m$^{-2}$ inland water area yr$^{-1}$ and, after multiplying per inland water area in each COSCAT, as g C per year. Finally, we estimated global OC burial by summing the total OC burial (as g C per year) in all COSCATs and, after dividing this sum by the global inland water area, we got the global average areal OC burial rate (g C m$^{-2}$ inland water yr$^{-1}$). We also calculated OC burial divided by watershed area (g C m$^{-2}$ land yr$^{-1}$) for each COSCAT, and for the total world land area.

The analyzes described above were performed four times by combining the two different methods for modeling (A and S) and the two inland water inventories (G and R), producing four global OC burial scenarios (see main text for details). In method S, however, these analyzes were applied only for COSCATs with average annual temperature <15 °C. The upscaling for ≥15 °C COSCATs was performed by using median OC burial values from all lakes and reservoirs of each size class (≥0.001–0.1 km$^2$, >0.1–1 km$^2$, >1–10 km$^2$, >10–100 km$^2$, >100 km$^2$) from our literature review. We used the median and not the mean to avoid overestimation caused by a few extremely high values, which is especially relevant for reservoirs.

From the average OC burial rates per watershed area (from our four scenarios) in the COSCATs, we estimated OC burial rates in the world's geographic zones, defined by the following latitude limits: southern temperate zone, from −66.3° to −40°; southern subtropical zone, from −40° to −23.3°; tropical zone, from −23.3° to 23.3°; northern subtropical zone, from 23.3° to 40°; northern temperate zone, from 40° to 66.3°; and northern polar zone, from 66.3° to 90°.

**Data availability**. The data on modern (up to ~150 years) OC burial rates in lakes and reservoirs, including geographic location, system characteristics and watershed features are given in the Supplementary Data.

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

## Acknowledgements

We thank N. Barros for the valuable discussions. The research leading to these results has received funding from the European Research Council under the European Union's Seventh Framework Programme (FP7/2007–2013)/ERC grant agreement n° 336642 to S.S. Additional financial support from the Swedish Research Council (VR) and the Swedish Research Council for Environment (Formas) to S.S. and from the Knut and Alice Wallenberg Foundation (grant KAW 2013.0091) to L.J.T. is acknowledged.

## Author contributions

R.M. and S.S. conceived and designed the study, and R.M. conducted most of the data analyzes and wrote the first draft of the paper together with S.S.; R.A.M. contributed with GIS analyzes, D.C. with statistical advise, and C.V. with expertize on lake and reservoir areas. P.R. contributed to study design and L.J.T. to data interpretation. S.S. contributed to statistical modeling. All authors contributed to writing the manuscript.

## Additional information

**Competing interests:** The authors declare no competing financial interests.

