## [Peer Review File · Nature Communications]

Reviewers' comments:

Reviewer #1 (Remarks to the Author):

The manuscript by Mendonca and others addresses carbon burial in lakes and reservoirs – a topic that has been of interest to the limnological community for a few decades and that will surely be of broader interest in the coming few years. Previous work on C burial in lakes and reservoirs has tended to be on only a few systems or on regions, Mendonca provides a much more representative sample of the diversity of the world's lakes and reservoirs by synthesizing data from hundreds of lakes in previously published work. A challenge for the broader scientific community is including C burial in lakes and reservoirs as part of global carbon budgets, because the processes governing C flux in the diversity of aquatic systems are not well quantified.

While the insights provided by this work are important, they are somewhat incremental. We have known for a long time that lakes and reservoirs are important sinks for C and of a magnitude similar to that of the oceans. However, Mendonca improves those estimates and provides additional interpretation by identifying important correlates. These correlates are then used to scale from the dataset to the globe. Although the work is empirical, it does provide insights that will be useful for process-based models.

Overall, this is a really nice paper that will likely inspire additional work by others to include lakes and reservoirs in global-scale C cycling models. The criticisms of this work are minor and are included in a few points below.

Paragraph beginning line 189 could be deleted, as it spins off into comparison of methane emissions, which really are out of scope here.

Final two paragraphs could be condensed into 1, considering most of the points had been made previously in the manuscript.

Discussion should address new global lakes datasets, like HYDROLAKE (Messinger et al), which could potentially improve these estimates.

Line 62: define COSCAT

Figure 1: add number of lakes/reservoirs either in the caption or on the figure [ie. Lake (n = #)]. Would benefit from outlining the circles and adding a transparent color so readers get a sense of how much data there are.

Figure 3: Again, would benefit from outlining the circles and adding a transparent color so readers get a sense of how much data there are.

Table 1: Please be consistent on use of commas and decimals in your tables.

Figure 7: The description of the x-axis is confusing

Figure 3 and Figure S6, have the same y-axis, but the log units are presented differently. My preference is the style shown in Fig 3, but either way, can they be consistent.

Reviewer #2 (Remarks to the Author):

This is an interesting paper that provides a new estimate of the annual burial of carbon in lakes and reservoirs world-wide. This study, like several that have been published in the past few years,

indicates that lakes and reservoirs provide a significant sink for organic carbon, with a carbon burial rate that is comparable to that of the world ocean. The authors have conducted a more thorough analysis than previous publications on carbon burial in lakes and reservoirs. The paper is clearly written for the most part, and merits publication in Nature Communications, after some of my concerns are addressed. I raise questions, edits and suggestions below:

Line 62: Define and explain what COSCATS is.

Lines 145-147: This sentence is confusing - lakes and reservoirs bury .15 PgC/yr whereas soils lose .15 PgC/yr. That is not a net global sequestration in soils, it is a net loss.

Line 185: 28-times

Lines 185-188: But methane lasts only 8 years in the atmosphere, then oxidizes to CO₂. Org C buried in sediment for more than 8 years, regardless of terrestrial or aquatic origin, has a negative net effect on radiative forcing, does it not?

Line 197: related to

Line 205: which suggests

Line 206: productivity turns

Lines 210-213: Soils are presently losing .15 PgC/yr, as you state largely due to agricultural practice. Assume for a moment that there was no agriculture. Would these undisturbed soils continually accumulate carbon or would they arrive at a steady-state concentration of C in the soil profile? I suspect the latter. Therefore lakes and reservoirs are more effective than soils in accumulating carbon and reservoirs should be considered a new C sink.

Table 1: This table is confusing. I now realize that the third line of data is the total of the two lines above it. But I was reading this line and the one below it as "all lakes." I suggest changing "all" to "total," and leaving a space between this line and the next one. Ditto for the "totals" in the lines below. And why does "all" on the 6th line of data not total the two numbers above it? Is it because on this line it is an average value for all lakes and reservoirs? If so, rename it "average for all."

Figure 2 is disturbingly patchy, showing a chaotic distribution of carbon burial, I guess because it reflects more than anything else the abundance of reservoirs around the world. Is this figure useful or misleading? I would relegate it to the SI, and instead insert your Supplemental Fig. S2 into the main text.

Speaking of Supplementary Figure S2, why do you not provide a linear fit for $T > 15^{\circ}\text{C}$? The linear fit for all data greatly over-estimates the warm water systems.

Finally, in looking over your data spreadsheet, I am surprised to find that you included data for only 9 of the ~30 largest lakes of the world, compiled by Allin and Johnson (2007). Why did you omit the data from 2/3 of the lakes they assessed?

Reviewer #3 (Remarks to the Author):

This is a well written paper with a comprehensive analysis. As a consequence my comments are relatively few. One aspect that was emphasized very clearly to me, however, was that the 'global lakes and reservoirs' used for analysis were very unevenly distributed. In fact there do not appear to be any data obtained from the SS and TS regions identified by the authors. In general the Asian-Pacific region is grossly under-represented. I am certainly aware of measurements from lake basins in these regions, so it is a little disappointing to see the absence of data from them. A subtle change in title may therefore be warranted, to 'Organic Carbon Burial in Lakes and Reservoirs Across the Globe'. The paucity of data from certain regions needs to be fully discussed in the main body of this paper. Detailed comments:

line 41: change fundamental to considerable.

line 62: COSCATS is only defined later.

line 63: the 'whole basin' initially confused me. These are evidently rates within the waterbody.

lines 83-84: catchment or catchments?

line 92: similarly to similar.

line 96, 99, 121 use a superscript for the squared term.

line 167 argumentation to argument.

line 185: 28-fold. Note the intermediate step of water column oxidation of CH₄ acting to reduce warming potential.

The last sentence of page 6 is really awkwardly worded. Please re-write.

line 205: what to which.

line 210: reduce to ...compensate for corresponding C emissions.

line 219: drier.

line 223: insert comma after Also.

Reference 32: use lower case.

Table 1: use decimal point rather than comma.

Supplementary material: there are lots of simply typographical errors in the supplementary material. These relate to superscripts, no space between numerical values and units, etc. Some good editing is required, with some specific examples being:

line 32: separate categories properly (<, ≤, etc.) *also lines 86-87 and 109).

lines 33-34: please clarify exactly lake basin slope vs. slope of the lake bed.

lines 38-40: would the two sites really be representative - some critical evaluation could be useful here and presumably two sites may have been selected to show different properties.

line 53: what are 1,000m maps?

line 104: this is the global OC (divided by watershed area)?

Reviewer #1 (Remarks to the Author):

The manuscript by Mendonca and others addresses carbon burial in lakes and reservoirs – a topic that has been of interest to the limnological community for a few decades and that will surely be of broader interest in the coming few years. Previous work on C burial in lakes and reservoirs has tended to be on only a few systems or on regions, Mendonca provides a much more representative sample of the diversity of the world's lakes and reservoirs by synthesizing data from hundreds of lakes in previously published work. A challenge for the broader scientific community is including C burial in lakes and reservoirs as part of global carbon budgets, because the processes governing C flux in the diversity of aquatic systems are not well quantified.

While the insights provided by this work are important, they are somewhat incremental. We have known for a long time that lakes and reservoirs are important sinks for C and of a magnitude similar to that of the oceans. However, Mendonca improves those estimates and provides additional interpretation by identifying important correlates. These correlates are then used to scale from the dataset to the globe. Although the work is empirical, it does provide insights that will be useful for process-based models.

Overall, this is a really nice paper that will likely inspire additional work by others to include lakes and reservoirs in global-scale C cycling models. The criticisms of this work are minor and are included in a few points below.

Paragraph beginning line 189 could be deleted, as it spins off into comparison of methane emissions, which really are out of scope here.

We are pleased that the reviewer expects our work will be useful to process-based models and may inspire additional work including lakes in global-scale C models. We realize that the paragraph starting at line 189 (now at line 198) was written in a way that may invoke the impression that it focusses on methane emissions. Our intention was to compare carbon burial with carbon (CO₂) emission, thereby putting the burial in the perspective of the overall C balance of lakes and reservoirs. To make this clearer, we rephrased this paragraph, including an initial sentence that clarifies its purpose (lines 198-200).

Final two paragraphs could be condensed into 1, considering most of the points had been made previously in the manuscript.

We agree that these paragraphs were to some extent repeating points made earlier in the manuscript, and we have now edited the text to avoid repetitions, moved some content to other places, and also included a point made by reviewer #2 – to make it clear that lakes and reservoirs tend to be more effective C sinks than soils. In this paragraph, we also want to draw special attention to the extent to which OC burial in inland waters represents a new C sink, a conceptually important issue that does not receive appropriate attention (see also reply to Reviewer #3).

The last paragraph is important for underlining the paucity of OC burial data from certain regions, and it also points towards the need for future studies. We now also mentioned HYDROLAKES (and other recent inland water area inventories) in this paragraph. We decided to keep the main content of last paragraph but edited it to eliminate any repetitive text, and we also opted to keep the last two paragraphs separated.

Discussion should address new global lakes datasets, like HYDROLAKE (Messinger et al), which could potentially improve these estimates.

We appreciate the suggestion to discuss the HYDROLAKES dataset in our manuscript. We agree that new inventories of inland water areas may help to improve estimates of carbon fluxes in inland waters. We now mention the HYDROLAKES dataset, but also other inventories of global lake area recently published (GIW v1.0 - Feng et al. 2000, and Pekel's time-resolved dataset- Pekel et al. 2016) in the last paragraph (lines 234-236).

- **Feng et al. A global, high-resolution (30-m) inland water body dataset for 2000: First results of a topographic–spectral classification algorithm. *International Journal of Digital Earth* 9, 113-133 (2016)**
- **Pekel et al. High-resolution mapping of global surface water and its long-term changes. *Nature* (2016)**

Line 62: define COSCAT

We now define it (lines 62-63).

Figure 1: add number of lakes/reservoirs either in the caption or on the figure [ie. Lake (n = #)]. Would benefit from outlining the circles and adding a transparent color so readers get a sense of how much data there are.

We made these modifications.

Figure 3: Again, should benefit from outlining the circles and adding a transparent color so readers get a sense of how much data there are.

We made this modification.

Table 1: Please be consistent on use of commas and decimals in your tables.

We checked for consistency throughout the ms.

Figure 7: The description of the x-axis is confusing

We rephrased it.

Figure 3 and Figure S6, have the same y-axis, but the log units are presented differently. My preference is the style shown in Fig 3, but either way, can they be consistent.

The purpose of the two plots is different. Fig. 3 is intended to illustrate the wide range of OC burial / CO₂ emission values and therefore the untransformed raw data are shown, albeit the axis is logged to improve visibility of the plotted data. Fig. S6 (now referred to as Supplementary Fig. 6), in contrast, shows a regression analysis, and transformation was an important step in data pre-treatment prior to regression analyses (see Methods, now presented in main text file). Accordingly, all regression results in the paper (both graphs and regression statistics) are reported based on log₁₀-transformed variables (if required). We therefore prefer not to change the style of these two graphs.

Reviewer #2 (Remarks to the Author):

This is an interesting paper that provides a new estimate of the annual burial of carbon in lakes and reservoirs world-wide. This study, like several that have been published in the past few years, indicates that lakes and reservoirs provide a significant sink for organic carbon, with a carbon burial rate that is comparable to that of the world ocean. The authors have conducted a more thorough analysis than previous publications on carbon burial in lakes and reservoirs. The paper is clearly written for the most part, and merits publication in Nature Communications, after some of my concerns are addressed. I raise questions, edits and suggestions below:

We appreciate that this reviewer finds that the manuscript is mostly clearly written and merits publication in Nature Communications.

Line 62: Define and explain what COSCATS is.

We now define it (lines 62-63)

Lines 145-147: This sentence is confusing - lakes and reservoirs bury .15 PgC/yr whereas soils lose .15 PgC/yr. That is not a net global sequestration in soils, it is a net loss.

We replaced “*sequestration in soils*” with “*flux of soils*” (line 151).

Line 185: 28-times

Following the suggestion by reviewer #3, we replaced “28-time” with “28-fold” (now at line 194).

Lines 185-188: But methane lasts only 8 years in the atmosphere, then oxidizes to CO₂. Org C buried in sediment for more than 8 years, regardless of terrestrial or aquatic origin, has a negative net effect on radiative forcing, does it not?

Timescale is indeed an important issue, that is why the global warming potential of various greenhouse gases is calculated at specific timescales. Considering the lifetime of CH₄ in the atmosphere of a bit over 8 years, at a 100 yr timescale CH₄ traps 28 times more heat in the atmosphere than the same amount of CO₂ (IPCC 2013). At a timescale of 20 years, the global warming potential of CH₄ would be even higher, 84 times that of CO₂. We now state explicitly

that our comparison is relevant at a 100 year interval only, which also is the relevant timescale for comparison with recent burial, as defined in our study (lines 193-197).

Line 197: related to

We made this correction.

Line 205: which suggests

We made this correction.

Line 206: productivity turns

We made this correction.

Lines 210-213: Soils are presently losing .15 PgC/yr, as you state largely due to agricultural practice. Assume for a moment that there was no agriculture. Would these undisturbed soils continually accumulate carbon or would they arrive at a steady-state concentration of C in the soil profile? I suspect the latter. Therefore lakes and reservoirs are more effective than soils in accumulating carbon and reservoirs should be considered a new C sink.

While it may be possible that soil C accumulation arrives at a steady state at some point of time, studies of the pre-Anthropocene C cycle estimate a C sink in soils, i.e. a continuous C accumulation (e.g. Regnier et al. 2013, Nature). At the same time, we do agree that lakes and reservoirs are probably more effective than soils in accumulating carbon (as evident from the same study, Regnier et al. 2013). We rephrased this part of the text, now stating that C burial in lakes and reservoirs is more effective than in soils, but also emphasizing and summarizing earlier made points (lines 164-171) that the degree to which inland water C burial can be accounted as a new C sink is currently unclear (lines 220-222).

Table 1: This table is confusing. I now realize that the third line of data is the total of the two lines above it. But I was reading this line and the one below it as "all lakes." I suggest changing "all" to "total," and leaving a space between this line and the next one. Ditto for the "totals" in the lines below. And why does "all" on the 6th line of data not total the two numbers above it? Is it because on this line it is an average value for all lakes and reservoirs? If so, rename it "average for all."

We replaced "All" with "Total". The values for "Total" are the sum of the values for lakes and reservoirs in all cases, except for the OC burial per water body area, which we now call "Average, Total". In this case, the total OC burial is calculated as the weighted average, considering the fluxes and the area occupied by lakes and reservoirs in each COSCAT. We have added an asterisk to that line and explained how the values were calculated as table footnote. We also increased the line thickness between the main row categories of the table to improve clarity.

Figure 2 is disturbingly patchy, showing a chaotic distribution of carbon burial, I guess because it reflects more than anything else the abundance of reservoirs around the world. Is this figure useful or misleading? I would relegate it to the SI, and instead insert your Supplemental Fig. S2 into the main text.

Figure 2 shows indeed a very patchy pattern, but these maps do show core results emerging from our analyses, and we believe it is important rather than misleading to show these results

and the patchiness to the readers. The main reasons for the patchiness are 1) OC burial data is highly influenced by the density of lakes and reservoirs in the COSCATs, which is also relatively patchy (see Supplementary Figure 4); 2) and OC burial data is still very scarce and highly variable (see range of values in Supplementary Figure 8). Both points are now explicitly discussed in the manuscript (lines 121-124, where we added the sentence “Large variability in OC burial rates is observed within each geographic zone (Fig. 2) due to the scarcity and uneven distribution of OC burial measurements (Supplementary Fig. 1) and to the relatively patchy distribution of inland waters (Supplementary Fig. 8)”).

Figure S2 (which we now refer to as Supplementary Fig. 2) shows the relationship between OC burial and temperature, which is not a main result but rather a part of the model development, so we prefer to keep it in the supplementary material.

Speaking of Supplementary Figure S2, why do you not provide a linear fit for $T > 15^{\circ}\text{C}$? The linear fit for all data greatly over-estimates the warm water systems.

The purpose of showing the linear fit for all data was exactly to show that the model including all data overestimates OC burial in warm regions; this was also clearly spelled out in the text (line 95-98). The two linear fits in Supplementary Figure 2 represent what is used in the two models we apply to the data (method A, “all”, and method S, “split”). As was stated in the manuscript (lines 102-104), no acceptable model could be fit to lakes $>15^{\circ}\text{C}$ because of scarcity of data from warm regions of the world.

Finally, in looking over your data spreadsheet, I am surprised to find that you included data for only 9 of the ~30 largest lakes of the world, compiled by Allin and Johnson (2007). Why did you omit the data from 2/3 of the lakes they assessed?

Because our paper was focused on modern OC burial rates (i.e. burial over the last ~150 years), we carefully checked the methods of all data reported in papers that we included in our literature review. In the case of Alin and Johnson (2007), most lakes were excluded from our dataset because (1) the data was from long-term (thousands of years or Holocene-scale) OC burial, or (2) we could not find the original reference and, thus, we could not determine the timescale considered, or (3) the source reference was already in our dataset.

Reviewer #3 (Remarks to the Author):

This is a well written paper with a comprehensive analysis. As a consequence my comments are relatively few. One aspect that was emphasized very clearly to me, however, was that the 'global lakes and reservoirs' used for analysis were very unevenly distributed. In fact there do not appear to be any data obtained from the SS and TS regions identified by the authors. In general the Asian-Pacific region is grossly under-represented. I am certainly aware of measurements from lake basins in these regions, so it is a little disappointing to see the absence of data from them. A subtle change in title may therefore be warranted, to 'Organic Carbon Burial in Lakes and Reservoirs Across the Globe'. The paucity of data from certain regions needs to be fully discussed in the main body of this paper.

We are pleased that also this reviewer finds that our manuscript is well written and merits only few comments. We fully agree that data paucity is an issue, particularly in some regions of the

world, and we therefore explicitly discuss data paucity throughout the entire manuscript. Nevertheless, after excessive search of the literature, we compiled the so-far largest collection of OC burial data from inland waters, and we develop models from those data which then are applied to the globe. We therefore think that keeping the title (“burial in global lakes and reservoirs”) is warranted, not the least since we explicitly deal with uncertainties in the global estimates by deriving ranges from the different scenarios. The reviewer’s title suggestion (“burial in lakes and reservoirs across the globe”) may give the impression that our study is a mere collection of empirical data, and would not adequately reflect the modeling and upscaling parts of our study.

While we cannot exclude that we may have missed some data during our search, there may be another reason to suspect that we are lacking data. There are many studies of inland water sediments within many fields of science (palaeostudies, contaminant science, organic geochemistry, etc), and while such studies often include data that could be used to calculate OC burial (chronologies, mass accumulation rates, sedimentation rates, OC content, density), these data are only rarely assembled to calculate and report OC burial, and instead dispersed over different papers published in different disciplines. Perhaps it is such studies the reviewer points towards in the Pacific-Asian region.

Detailed comments:

line 41: change fundamental to considerable.

We made this modification.

line 62: COSCATS is only defined later.

We moved the definition to this sentence.

line 63: the 'whole basin' initially confused me. These are evidently rates within the waterbody.

We replaced “*whole-basin*” with “*whole-system*” as we use the word “*system*” along the manuscript to refer to lakes and reservoirs.

lines 83-84: catchment or catchments?

We replaced “*catchment*” with “*catchments*”.

line 92: similarly to similar.

We made this correction.

line 96, 99, 121 use a superscript for the squared term.

We made this correction, and we replaced the R^2 values with the adjusted R^2 values, as in Table S1.

line 167 argumentation to argument.

We made this correction.

line 185: 28-fold. Note the intermediate step of water column oxidation of CH₄ acting to reduce warming potential.

We replaced “28-times” with “28-fold”. We now also mention that some of the CH₄ can be microbially oxidized to CO₂ (lines 192-193).

The last sentence of page 6 is really awkwardly worded. Please re-write.

We edited the sentence, splitting it into two shorter sentences.

line 205: what to which.

We made this correction.

line 210: reduce to ...compensate for corresponding C emissions.

We made this modification.

line 219: drier.

We made this correction.

line 223: insert comma after Also.

We made this correction.

Reference 32: use lower case.

We made this correction.

Table 1: use decimal point rather than comma.

We made this correction.

Supplementary material: there are lots of simply typographical errors in the supplementary material. These relate to superscripts, no space between numerical values and units, etc. Some good editing is required, with some specific examples being:

We have thoroughly revised the supplementary material.

line 32: separate categories properly (<, ≤, etc.) *also lines 86-87 and 109).

We made this correction.

lines 33-34: please clarify exactly lake basin slope vs. slope of the lake bed.

We made this correction.

lines 38-40: would the two sites really be representative - some critical evaluation could be useful here and presumably two sites may have been selected to show different properties.

Unless measured, spatial representative values can only be estimated, and there are no reasons to assume that one of the two approaches that we used (a and b) is superior to the other. Approach (a) uses fixed average sediment focusing factors binned by categories of maximum depth to get a spatially representative value from one measurement of OC burial, and this averaging necessarily induces uncertainty. Approach (b) uses averaging of OC burial measurements in lakes/reservoirs with several measurements of OC burial, and as the reviewer correctly points out, this may not be returning a spatially representative value depending on how the sites in the original study were selected. Since none of the two approaches seems superior to the other, a comparison is futile. We revisited the data and found that approach (b) was only applied on 4 systems, and only in 1 system, there were 2 OC burial measurements only; in the remaining 3 systems, approach (b) was used with 4, 4 and 8 OC burial measurements each. None of these 4 sites was deviating statistically from the rest of the data. We conclude that the overall effect of potential uncertainties associated with approach (b) on our analysis probably is small, and we have added corresponding text to the supplementary information (lines 260-264).

line 53: what are 1,000m maps?

This is the map resolution which means that the pixels (the smallest feature that can be represented in that map) measures 1,000 x 1,000 meters. This has been clarified in the text (line 285 of main text file).

line 104: this is the global OC (divided by watershed area)?

Yes, this is the global OC burial (in g C yr⁻¹) divided by the world's land area, and we calculated it for each COSCAT as well. This is now clarified in the text (lines 340-341 of main text file).

Reviewers' comments:

Reviewer #1 (Remarks to the Author):

The authors provided acceptable and useful responses to criticisms and made appropriate updates to the manuscript. This is an impressive piece of work and as stated previously an important contribution to the field. I have no additional criticism of the ideas and work; however, I do have a number of recommendations for grammatical corrections and changes of clarification.

Line 33: Change "...higher share..." to "...higher proportion..."

Line 59: Change "...data is..." to "...data are..."

Line 69: Change "...at a higher rates..." to "...at higher rates..."

Line 70: What are the better conditions for OC preservation? Presumably, these are cold and anoxic conditions.

Paragraph beginning line 142: There are a couple of uses of 'which however' that I find distracting.

Line 152: The use of 'but' after the first half of the sentence implies negation of the first half. It would be better to rephrase similarly to, "...represents a transfer of 0.3% (range of 0.1% to 0.5% for the four scenarios) of global terrestrial net primary production (~52 Pg C yr⁻¹ 152, estimate for 1990-2009, IPCC) to sinks operating at timescales of decades-centuries (reservoirs) to millennia (lakes)."

Line 163: The literature cited provide more than implication. Consider changing this sentence to, "...oxygen in bottom waters are proposed mechanisms by which reservoirs accumulate OC..."

Line 164: Remove 'in'.

Line 168: The use of 'however' is distracting.

Line 171: The phrasing of this long sentence is awkward.

Line 188: This sentence would be more clear if, "The burial of aquatic OC..." were changed to, "The burial of autochthonous OC..."

Line 223: I think the statement that C burial in lakes and reservoirs is more effective than in soils is vague enough that it is misleading because it could be interpreted as lakes and reservoirs being the main OC repository in the terrestrial biosphere. I assume what the authors meant was that per unit area, lakes and reservoirs have higher burial rates. This can be clarified with a simple fix.

Reviewer #2 (Remarks to the Author):

The authors addressed to my satisfaction the points I raised in my review of the original submission. Now that I have read the revision, a few new concerns arise in my mind, which perhaps reflect a lack of understanding on my part. I apologize for not raising the following points in my original review.

I think I understand the basic 4-way split among AR, SR, AG, and SG approaches to arriving at global estimates of carbon burial in lakes and reservoirs. However the paper does not provide the

empirical relationships that link parameters of mean temperature, lake area, water depth, land cover, etc. that yield the mean carbon burial rate assigned to each COSCAT. I do not see them in the data spreadsheet that was provided, or in the supplemental material. Such information must be made available if this paper is to be published.

Without having a better understanding of the form of these relationships, I would guess that the only difference in OC burial between AG and SG approaches would be in COSCATs that have mean annual temperatures warmer than 15°C. Ditto between AR and SR approaches. And yet Fig 2 displays different OC burial values for "A" and "S" approaches in relatively high latitude COSCATs, such as Greenland and Siberia. Why would these regions yield different OC burial rates when temperatures in these areas fall below 15°C?

Given that warm (>15° C) regions have lower OC burial rates than predicted by the linear fit through data from lakes in all regions, I would expect the AR and AG approaches to always yield higher OC burial values than would the SR and SG approaches. Yet, in Table 1, SG reservoir values are higher than AG reservoir values – why?

The data spreadsheet should indicate what sediment focusing factor was used for those lakes where it was employed. Many of the columns in the data spreadsheet have way too many significant figures beyond the decimal point – unwarranted.

Finally, in retrospect, I believe the authors took the wrong approach when they decided to arrive at mean values of OC burial for COSCAT regions. Some of the larger regions stretch across widely disparate climate and vegetation zones. The Nile drainage, for example, extends from tropical equatorial Africa through the Sahara desert. How does one arrive at a reasonable average value for that? The Mississippi drainage encompasses the relatively humid eastern U.S. and the semi- arid high plains. The authors' next study should perhaps use the same techniques, but split the globe according to climatic or vegetation zones, where one might expect to find a simpler pattern, or trend, in OC burial rates.

Reviewer #3 (Remarks to the Author):

This is a re-review of the manuscript that I reviewed previously. I am satisfied that all points that I raised in the first instance have now been addressed, with appropriate responses where this is not the case, and I have nothing further to point out.

Reviewer #1 (Remarks to the Author):

The authors provided acceptable and useful responses to criticisms and made appropriate updates to the manuscript. This is an impressive piece of work and as stated previously an important contribution to the field. I have no additional criticism of the ideas and work; however, I do have a number of recommendations for grammatical corrections and changes of clarification.

We are pleased that the reviewer finds that the manuscript is an important contribution to the field.

Line 33: Change "...higher share..." to "...higher proportion..."

Done.

Line 59: Change "...data is..." to "...data are..."

Done.

Line 69: Change "...at a higher rates..." to "...at higher rates..."

Done.

Line 70: What are the better conditions for OC preservation? Presumably, these are cold and anoxic conditions.

Yes, it is mainly due to anoxic conditions. We now explained it in the text.

Paragraph beginning line 142: There are a couple of uses of 'which however' that I find distracting.

We replaced "which however" with "which" in that paragraph.

Line 152: The use of 'but' after the first half of the sentence implies negation of the first half. It would be better to rephrase similarly to, "...represents a transfer of 0.3% (range of 0.1% to 0.5% for the four scenarios) of global terrestrial net primary production (~52 Pg C yr⁻¹ 152, estimate for 1990-2009, IPCC) to sinks operating at timescales of decades-centuries (reservoirs) to millennia (lakes)."

We rephrased the sentence following the reviewer's suggestion.

Line 163: The literature cited provide more than implication. Consider changing this sentence to, "...oxygen in bottom waters are proposed mechanisms by which reservoirs accumulate OC..."

We rephrased the sentence.

Line 164: Remove 'in'.

Done.

Line 168: The use of 'however' is distracting.

We rephrased the sentence.

Line 171: The phrasing of this long sentence is awkward.

We rephrased this part of the text and split it in two sentences.

Line 188: This sentence would be more clear if, “The burial of aquatic OC...” were changed to, “The burial of autochthonous OC...”

Done.

Line 223: I think the statement that C burial in lakes and reservoirs is more effective than in soils is vague enough that it is misleading because it could be interpreted as lakes and reservoirs being the main OC repository in the terrestrial biosphere. I assume what the authors meant was that per unit area, lakes and reservoirs have higher burial rates. This can be clarified with a simple fix.

We agree that the sentence was misleading and we fixed it.

Reviewer #2 (Remarks to the Author):

The authors addressed to my satisfaction the points I raised in my review of the original submission. Now that I have read the revision, a few new concerns arise in my mind, which perhaps reflect a lack of understanding on my part. I apologize for not raising the following points in my original review.

We are glad that the reviewer considers our responses to the points raised in the first revision of our manuscript as satisfactory.

I think I understand the basic 4-way split among AR, SR, AG, and SG approaches to arriving at global estimates of carbon burial in lakes and reservoirs. However the paper does not provide the empirical relationships that link parameters of mean temperature, lake area, water depth, land cover, etc. that yield the mean carbon burial rate assigned to each COSCAT. I do not see them in the data spreadsheet that was provided, or in the supplemental material. Such information must be made available if this paper is to be published.

We did show the effect of each parameter on the prediction of OC burial by Models 1 and 2 in Supplementary Table 1. The reference to this table was given the main text (now lines 85, 99 and 102).

Without having a better understanding of the form of these relationships, I would guess that the only difference in OC burial between AG and SG approaches would be in COSCATs that have mean annual temperatures warmer than 15°C. Ditto between AR and SR approaches. And yet Fig 2 displays different OC burial values for “A” and “S” approaches in relatively high latitude COSCATs, such as Greenland and Siberia. Why would these regions yield different OC burial rates when temperatures in these areas fall below 15°C?

Not only the effect of temperature on OC burial is different between the A and S approaches. Also the effects (i.e. regression slopes) of all other predictors on OC burial were different between the two models at the base of the A and S approaches (see Supplementary Table 1),

because the models were built based on different datasets. For this reason, the models result in different OC burial rates in COSCATs colder than 15°C as well.

Given that warm (>15° C) regions have lower OC burial rates than predicted by the linear fit through data from lakes in all regions, I would expect the AR and AG approaches to always yield higher OC burial values than would the SR and SG approaches. Yet, in Table 1, SG reservoir values are higher than AG reservoir values – why?

In the S approach, we used a fixed value, the median OC burial rate per area class, for COSCATs >15°C. The median values of OC burial rates (per area class) for reservoirs in COSCATs > 15°C, were higher than what Model 1 (used in A scenarios) predicted for reservoirs in most >15°C COSCATs. Consequently, in >15°C COSCATs where reservoirs represent a significant area, the “S” approach resulted in overall higher OC burial than the “A” approach.

The data spreadsheet should indicate what sediment focusing factor was used for those lakes where it was employed. Many of the columns in the data spreadsheet have way too many significant figures beyond the decimal point – unwarranted.

We added a column with the sediment focusing factor applied to the Supplementary Data table and we adjusted the decimals.

Finally, in retrospect, I believe the authors took the wrong approach when they decided to arrive at mean values of OC burial for COSCAT regions. Some of the larger regions stretch across widely disparate climate and vegetation zones. The Nile drainage, for example, extends from tropical equatorial Africa through the Sahara desert. How does one arrive at a reasonable average value for that? The Mississippi drainage encompasses the relatively humid eastern U.S. and the semi-arid high plains. The authors’ next study should perhaps use the same techniques, but split the globe according to climatic or vegetation zones, where one might expect to find a simpler pattern, or trend, in OC burial rates.

We appreciate the reviewer’s suggestion and we agree that some of the COSCATs are very large and heterogeneous, implying that OC burial probably varies a lot within them. However, it is likely that also other classifications inland water systems (e.g. by vegetation zone, or by climate zone) would have the problem of within-class variability in parameters that affect OC burial. We believe that the best way to improve the burial estimates in the future would be to use finer spatial scale.

Reviewer #3 (Remarks to the Author):

This is a re-review of the manuscript that I reviewed previously. I am satisfied that all points that I raised in the first instance have now been addressed, with appropriate responses where this is not the case, and I have nothing further to point out.

We are pleased that the reviewer is satisfied with the way we addressed the points raised about our manuscript.

Reviewers' Comments:

Reviewer #2:

Remarks to the Author:

The authors' rebuttal cleared up many of the concerns I expressed in my second review of the manuscript. There are just a few more dots to connect --- . The caption of Supplementary Table 1 should explicitly state the form of the equation to be used with the tabulated numbers to arrive at an organic carbon burial rate. I presume that the intercept and the "Lake or reservoir type" numbers are additive constants, while the other numbers are numerical coefficients in the equation?

Regarding the linear equations in the caption of Supplementary Fig 2 - shouldn't the slopes of these equations be positive?

I disagree with the authors' rebuttal to my concern about using average burial rates across COSCATs rather than across climatic or vegetational zones - they may find more in the community who share my view on this. Nevertheless, I would not insist on their undertaking this revised approach prior to acceptance of this paper. Something to do in the future!

Reviewer #2 (Remarks to the Author):

The authors' rebuttal cleared up many of the concerns I expressed in my second review of the manuscript. There are just a few more dots to connect --- . The caption of Supplementary Table 1 should explicitly state the form of the equation to be used with the tabulated numbers to arrive at an organic carbon burial rate. I presume that the intercept and the "Lake or reservoir type" numbers are additive constants, while the other numbers are numerical coefficients in the equation?

As recommended, we added the equation to the table caption.

Regarding the linear equations in the caption of Supplementary Fig 2 - shouldn't the slopes of these equations be positive?

Yes, the slopes are positive. The negative signs were a typing error which we now fixed. We appreciate that the reviewer noticed it.

I disagree with the authors' rebuttal to my concern about using average burial rates across COSCATs rather than across climatic or vegetational zones - they may find more in the community who share my view on this. Nevertheless, I would not insist on their undertaking this revised approach prior to acceptance of this paper. Something to do in the future!

We agree with the reviewer that future studies should investigate in further detail in how far different strategies for upscaling arrive at different results.